# The Role of Adjuvant Chemotherapy in pN1 (IIB/IIIA) NSCLC Patients Who Undergo Pneumonectomy: Is It Still Justified in the Modern Era?

**DOI:** 10.3390/cancers16173041

**Published:** 2024-08-31

**Authors:** Antonio Mazzella, Riccardo Orlandi, Sebastiano Maiorca, Clarissa Uslenghi, Patrick Maisonneuve, Monica Casiraghi, Luca Bertolaccini, Lorenzo Spaggiari

**Affiliations:** 1Division of Thoracic Surgery, IEO, European Institute of Oncology IRCCS, Via Ripamonti, 435, 20141 Milan, Italy; riccardo.orlandi@unimi.it (R.O.); sebastiano.maiorca@unimi.it (S.M.); clarissa.uslenghi@unimi.it (C.U.); monica.casiraghi@ieo.it (M.C.); luca.bertolaccini@gmail.com (L.B.); lorenzo.spaggiari@ieo.it (L.S.); 2Division of Epidemiology and Biostatistics, IEO, European Institute of Oncology IRCCS, 20141 Milan, Italy; patrick.maisonneuve@ieo.it; 3Department of Oncology and Haemato-Oncology, University of Milan, 20141 Milan, Italy

**Keywords:** pneumonectomy, NSCLC, lung cancer, chemotherapy, adjuvant treatment, N1, lymph node involvements

## Abstract

**Simple Summary:**

We aimed to evaluate the role of adjuvant chemotherapy in patients who undergo pneumonectomy for pN1 NSCLC. We found no significant differences after 5 years between two cohorts (adjuvant CT and no adjuvant CT) in terms of the overall survival (*p*: 0.31), cancer-specific survival (*p*: 0.59), disease-free survival (*p*: 0.94), and relapse rate (*p*: 0.76). Patients with pN1 NSCLC that was completely resected through pneumonectomy and radical lymphadenectomy may represent a particular cohort, which could be strictly followed up without adjuvant chemotherapy.

**Abstract:**

Introduction: We aimed to assess our 25-year experience in order to evaluate the role of adjuvant chemotherapy in patients who undergo pneumonectomy for pN1 NSCLC. Materials and Methods: We retrospectively reviewed the outcomes and medical records of patients undergoing pneumonectomy for NSCLC with pathological diagnosis of pN1, excluding all patients who underwent neoadjuvant treatment. We compared patients treated with adjuvant chemotherapy with patients who did not undergo neoadjuvant treatment during a follow-up soon after surgery. Gray’s test was used to assess differences in the cumulative incidence of relapse or CSS between the different groups. Kaplan–Meier methods were used for drawing overall survival (OS) plots. In order to assess differences in survival between the groups, the log-rank test was used. The cumulative incidence of relapse, CSS, and OS were calculated at 1, 2, 3, 4, and 5 years of follow-up. Results: The 30-day and 90-day mortality rates of our cohort were 6% and 11,6%. Excluding the first three months after surgery (deaths linked to postoperative comorbidity), after 5 years we found no significant differences between the two cohorts (adjuvant CT and no adjuvant CT) in terms of the overall survival (OS) (*p*: 0.31), cancer-specific survival (CSS) (*p*: 0.59), disease-free survival (DFS) (*p*: 0.94), and relapse rate (*p*: 0.76). Conclusions: Patients with pN1 NSCLC that was completely resected through pneumonectomy and radical lymphadenectomy may represent a particular cohort, which could be strictly followed up without adjuvant chemotherapy.

## 1. Introduction

In the last 20 years, we have witnessed a very important improvement in surgical techniques, which have allowed us to perform parenchymal-sparing intervention (i.e., bronchial or vascular sleeve) even with minimally invasive methods.

Aside from this last consideration, pneumonectomy remains the only surgical therapeutic alternative to obtaining oncologic radicality in some cases. Nevertheless, this operation is universally recognized as presenting high postoperative morbidity [1,2,3] and mortality rates (5–9% among all pulmonary resections) [4,5,6,7].

A standard regimen of adjuvant chemotherapy has long been considered an excellent tool in the multimodal treatment of stage II-III NSCLC, though with limited gains of about 5.4% in overall survival (OS) and 5.8% in disease-free survival (DFS), as highlighted some time ago by the well-known LACE study [8]. Since that time, many authoritative studies and trials have confirmed these results for stage IIB-IIIA NSCLC [9,10,11,12,13,14]. In these studies, overall survival is always improved in patients who undergo adjuvant treatments, reaching percentages of 8–10% in some cases. In addition, the adjusted risk of death is significantly reduced in patients subjected to chemotherapy compared to controls [9].

The latest NCCN guidelines [NCCN] suggest that patients with completely resected IIB-IIIA (N1) NSCLC should undergo adjuvant systemic therapy with either Osimertinib—if EGFR-mutated and after receiving standard adjuvant chemotherapy, or if ineligible to receive platinum-based chemotherapy—or Atezolizumab, if PD-L1 ≥ 1% and EGFR and ALK are both the wild type, and after receiving standard adjuvant chemotherapy (www.nccn.org/guidelines, accessed on 7 July 2024).

It has long been considered standard practice to administer adjuvant treatment (AT) to N1 patients, despite the survival benefit being approximately 5–8%, as extensively demonstrated in the literature. However, the patients enrolled in the series or trials considered so far largely underwent lobar resections and only a small cohort received pneumonectomies. It is a well-known fact that patients undergoing pneumonectomy are less likely to receive chemotherapy, mainly due to greater incidence of postoperative complications, which prevent them from starting adjuvant chemotherapy.

Therefore, the exact role of chemotherapy in the treatment of stage II-III NSCLC pN1 patients undergoing pneumonectomy has never been extensively and conclusively investigated.

We aimed to assess our 25-year experience at the European Institute of Oncology, in order to evaluate the role of adjuvant chemotherapy in patients who undergo pneumonectomy for pN1 NSCLC.

## 2. Materials and Methods

### 2.1. Patients

We retrospectively reviewed, according to the STROBE (Strengthening the Reporting of Observational Studies in Epidemiology) statement, our single-centre experience [15]. We retrospectively reviewed pre, peri, and postoperative outcomes (Table 1). Written informed consent to undergo the procedure and for the use of clinical imaging data for scientific or educational purposes, or both, was obtained from all patients before surgery. The study was approved by the Ethics Committee of the European Institute of Oncology (UID 4436).

A total of 929 patients were treated for lung cancer through pneumonectomy between 1998 and 2023 at the European Institute of Oncology; among these, we selected 334 patients (38.8%) who underwent pneumonectomy for NSCLC with pathological diagnosis of pN1. We excluded all patients who underwent neoadjuvant treatment (Figure 1). Our final cohort consisted of 189 patients (153 M, 36 F).

Specifically, we selected patients who, according to the 8th edition of TNM, were affected by pathological stage IIB (T1a-2b N1)/IIIA (T3-4 N1) NSCLC, without neoadjuvant treatment.

Preoperative staging included a total body computed tomography scan (CT-scan), positron emission tomography (PET) with fluorodeoxyglucose (FDG), cardiological examination, and a pulmonary function test. We added perfusional pulmonary scintigraphy and a cardiopulmonary exercising test (CPET) as routine exams from 2017. All the exams were performed in the 5 weeks before surgery.

Surgical indication, operability, and following management were decided during multidisciplinary meetings of thoracic surgeons and oncologists

We performed pneumonectomy via lateral thoracotomy and a radical mediastinal lymph node dissection in all patients (station 4R, 7, 8, 9, 10 on the right; 5, 6, 7, 8, 9, 10 on the left).

After surgery, patients were immediately extubated in the operating or recovery room, and then transferred to the thoracic department or to the intensive care unit (ICU). Whenever the extubation was not possible, due to anesthesiology or respiratory issues, the double-lumen tube was substituted by a 1-lumen tube. The patient was transferred to the ICU, and conventional mechanical ventilation was performed (tidal volume 8 mL/kg, PEEP 5 cm H_2_O). The pleural cavity was routinely drained; chest tubes were discharged on the second postoperative day, after stabilization of the mediastinum.

### 2.2. Indications for Postoperative Management/Treatment

Decisions about postoperative treatments in our institution are made through individually based discussions during weekly Multidisciplinary Tumor Board meetings. On the basis of several guidelines, our board always suggests postoperative chemotherapy for all patients with pN1 and stage IIB (T1a-2b N1)/IIIA (T3-4 N1), except for patients with pre- or postoperative comorbidities contraindicating chemotherapy. The timing of the start of adjuvant treatment is 30–40 days after surgery.

In our cohort, 77 patients received postoperative chemotherapy (40.7%), consisting of a doublet of platinum and pemetrexed if possible. In 96 cases out of 189 (50.8%), no adjuvant treatment was adopted because of the patient’s postoperative clinical condition or comorbidities. In 16 cases (8.4%), postoperative radiotherapy was performed in order to treat R1 pathological tissue after surgery (on the bronchus or on the chest wall).

In the last 12 months, in accordance with the results of the ADAURA study [16] and the latest NCCN guidelines, we have required different molecular assets to treat surgical specimens with EGFR and ALK mutations, as well as different dosages of PD-L1 in case of adenocarcinoma and PD-L1 in case of squamous cells. Further investigations are mandatory in order to propose tailor-made treatments on the basis of biologic assessment. Particularly, in the case of EGFR-mutated adenocarcinoma, we propose an adjuvant EGFR TKI (thyroxin-kinase inhibitor) therapy; in the case of overexpression of PD-L1 (>50%), we propose a combined CT-immunotherapy.

### 2.3. Patient Follow-Up

Patients underwent clinical examination every month in the first year. Follow-up included a physical examination, chest X-ray, blood chemistry analysis, tumor blood marker analysis, and a computed tomography scan (every 3 months in the first year, every 6 months in the following years).

### 2.4. Statistical Analysis

The cumulative incidence of relapse and cancer-specific survival (CSS) were estimated using the method of Kalbfeisch and Prentice, accounting for competing events (death as the first event for relapse, and postoperative death or death from another cause for cancer-specific survival). Gray’s test was used to assess differences in the cumulative incidence of relapse or CSS between groups. Overall survival (OS) plots were drawn using the Kaplan–Meier method, and the log-rank test was used to assess differences in survival between the groups. The cumulative incidence of relapse, CSS, and OS were assessed at 1, 2, 3, 4, and 5 years of follow-up. Hazard ratios (HRs) for the development of relapse or death were determined using univariate Cox proportional hazards regression models and multivariable models, adjusted per stage. Statistical analyses were performed using SAS software (version 9.4, SAS Institute, Cary, NC, USA). Statistical significance was defined as 2-sided *p* < 0.05.

## 3. Results

We performed 111 left and 78 right pneumonectomies in the examined period. Postoperative pathologic results showed 121 adenocarcinomas (64%), 63 squamous cell carcinomas (33.3%), 3 adenosquamous carcinomas (1.6%), and 2 undifferentiated tumor NSCLC (1.1%). Because of tumor positioning less than 2 cm away from the tracheal carina, tracheal sleeve pneumonectomy was performed in seven cases. All clinical and demographic data are expressed in Table 1.

A total of 96 patients out of 189 (50.8%) did not receive any adjuvant treatment. Of the rest of the patients (93 out of 189–49.8%), 73 received adjuvant chemotherapy; in 16 cases, only radiotherapy for R1 disease (microinvasion of resected bronchus or microinvasion of the parietal pleura/chest wall) was administered; in the other 4 cases, patients received adjuvant radio-chemotherapy (Table 1).

The 30-day mortality was 6% (13 patients), and 90-day mortality was 11.6% (22 patients). Among these, only two patients received adjuvant treatment. The other 20 patients did not receive any adjuvant treatment for their clinical postoperative conditions or postoperative complications (Table 2).

### 3.1. Overall Survival

The overall survival of the untreated population was 54%. The median OS was 30 months (0–229 months). The mean OS was 45 months. If we consider the time of intervention as time zero of observation, there is a significant difference (*p* = 0.007) between patients receiving overall adjuvant treatments (CT/RT/CT + RT) (62%) or not (46%) and between patients receiving adjuvant chemotherapy alone (64%) or not (47%), *p*: 0.01 (Figure 2).

However, we performed a second analysis to homogenize our cohort, considering “time zero” to be the 90-day period immediately following surgery (i.e., the critical period for surgical complications after pneumonectomy). So, we excluded the 22 patients who died within the first 3 months after surgery; indeed, in this case, the deaths were linked to severe postoperative complications or surgery-related problems. In this case, the overall survival was 60.5%, but there were no differences concerning survival (*p*: 0.48) between patients receiving adjuvant treatment (63%) or not (58%). We found the same results after 10 years (*p*: 0.71). Similarly, considering adjuvant chemotherapy alone, we found no significant difference between two cohorts (adjuvant CT: 65%, no adjuvant CT: 57%, *p*: 0.31) after 5 years or after 10 years (*p*: 0.32) (Figure 2 and Figure 3).

### 3.2. Cancer-Specific Survival

Regarding cancer-specific 5-year survival (considering time zero to be the time of the intervention), we found no significant difference either with regard to adjuvant treatments (no adjuvant treatments: 62%, adjuvant treatments: 68%, *p*: 0.58) or with regard to adjuvant chemotherapy alone (no adjuvant CT: 60%, adjuvant CT: 67%, *p*: 0.59). We observed the same results, excluding patients dying in the first 90 days after surgery (*p*: 0.31). The 10-year follow-up confirmed these results (*p*: 0.38) (Figure 2 and Figure 3).

### 3.3. Disease-Free Survival

Likewise, 5-year disease-free survival was significantly higher in the patients receiving adjuvant treatments (CT/RT/CT + RT) (50%) compared to those who did not receive adjuvant CT (41%), *p*: 0.07. Excluding the first three months after surgery, there are no differences between the two populations (no adjuvant treatments: 51%, adjuvant treatments: 52%, *p*: 0.94). The latter results are in accordance with the data after the 10-year follow-up (*p*: 0.94) (Figure 4).

Concerning the patients who only received adjuvant chemotherapy, the 5-year disease-free survival was significantly higher in the patients receiving adjuvant CT (53%) compared to those who did not receive it (40%), *p*: 0.03. Excluding the first three months after surgery, there are no differences between the two populations (no adjuvant CT: 49%, adjuvant CT: 55%, *p*: 0.48). We found the same results after the 10-year follow-up (*p*: 0.94) (Figure 5).

### 3.4. Relapse

The relapse rate at 5 years was 25.5%. No significant differences (*p*: 0.76) were found between patients receiving adjuvant chemotherapy (29 out of 77–27%) or not (34 out of 112–24%) (Figure 6).

## 4. Discussion

Several meta-analyses and randomized controlled trials have proven that adjuvant treatment after the resection of stage II-IIIA NSCLC could improve survival rate [8,9,10,11,12,13,14,17,18].

The latest NCCN guidelines [NCCN] suggest that completely resected IIB-IIIA (N1) NSCLC patients should undergo adjuvant systemic therapy with Osimertinib, if EGFR-mutated and after receiving standard adjuvant chemotherapy or if ineligible to receive platinum-based chemotherapy, or with Atezolizumab if PD-L1 ≥ 1%, EGFR and ALK are both the wild type, and standard adjuvant chemotherapy has been received.

A standard regimen of adjuvant chemotherapy has long been considered an important tool in the multimodal treatment of stage II–III NSCLC, though with limited gains of approximately 5.4% in OS and 5.8% in DFS. A post hoc analysis of the LACE report has suggested greater than expected OS and DFS rates of 20.4% and 16.3%, respectively, in pN1 NSCLC patients [18]. Several studies have confirmed this increased OS improvement in pN1-resected patients, with gains reaching up to 28.9% after 5 years for OS and 22.7% for DFS [14].

However, the other emerging aspect from these data is that the enrolled patients (NSCLC N1-resected patients) had largely undergone lobar resections and only a small cohort received pneumonectomies. Indeed, in the study demonstrating the most substantial overall survival (OS) advantage [14] for N1 patients undergoing adjuvant therapy (i.e., 28.9%), patients who underwent pneumonectomy were even excluded from the study. Patients undergoing pneumonectomy are less likely to receive chemotherapy, mainly due to greater incidence of postoperative complications, which prevent them from starting adjuvant chemotherapy.

Alam and colleagues [19] showed that patients undergoing pneumonectomy have markedly lower compliance to adjuvant therapy than those undergoing lesser resections, due to the higher number of toxicities experienced by the former.

Indeed, the results of chemotherapy after pneumonectomy have been disputed and its employment has long been debated due to the increased risk of impairment of the lung alveolocapillary membrane, which could contribute to lung damage, adding to post-pneumonectomy compensation phenomena and leading to respiratory functional impairment [20]. Therefore, the exact role of chemotherapy in the treatment of stage II-III NSCLC pN1 patients undergoing pneumonectomy has never been extensively and conclusively investigated. According to our results, patients undergoing pneumonectomy for NSCLC pN1 as well as adjuvant treatment may not have a significant advantage over those undergoing pneumonectomy without adjuvant treatment.

Chemotherapy could increase the risk of postoperative respiratory morbidity; in patients undergoing pneumonectomy, any event affecting the residual lung could be fatal. It has been suggested that the underlying mechanism of therapy-related damage to the lung is linked to the decrease in the alveolocapillary membrane diffusion capacity, which would amplify the damage caused by inflammatory agents during the perioperative course, facilitating intra-alveolar exudates [21].

If we consider pneumonectomy as a procedure inherently carrying a high rate of postoperative complications or respiratory comorbidities, with a mortality rate of around 5–9% [7,21,22,23,24] at 30 days, it goes without saying that an invasive medical treatment like chemotherapy could clearly exacerbate this risk. Wang et al. [25] reported their experience regarding the role of adjuvant chemotherapy after pneumonectomy for NSCLC. In their observation, adjuvant CT did not impact the overall survival (OS) of pneumonectomized patients; conversely, adjuvant CT represented an important risk factor for developing respiratory complications. Particularly, respiratory failure was one of the main causes of mortality in patients who underwent right pneumonectomy following adjuvant chemotherapy. On the other hand, adjuvant chemotherapy significantly improved the survival rate of patients who underwent left, but not right, pneumonectomy.

Concerning the association between chemotherapy, pneumonectomy, and N1 NSCLC, Speicher et al. affirmed that induction chemotherapy for cN1 NSCLC is not associated with improved survival [26], but induction CT may be rational in N1 patients in order to avoid pneumonectomy or to improve resectability. Broderick et al. [27] are on the same page, demonstrating that the survival of patients affected by stage IIIA NSCLC is not improved by neoadjuvant chemoradiation followed by pneumonectomy, compared to pneumonectomy followed by adjuvant chemoradiation.

Liang and colleagues investigated the early and long-term survival outcomes of patients undergoing pneumonectomy followed by adjuvant treatment compared to those undergoing pneumonectomy who were followed up with, and found that adjuvant treatment improved short-term survival but did not influence long-term outcomes, reporting a 5-year survival rate of 34% in the adjuvant treatment group vs. 21% in the non-adjuvant treatment group [28].

Therefore, if on the one hand the advantages of chemotherapy are still a point of debate, on the other hand, the risks of undertaking a systemic therapy such as chemotherapy must be considered for patients undergoing pneumonectomy. Recently, Brunelli and colleagues [29] analyzed the ESTS database to investigate the morbidity and mortality of pneumonectomy after neoadjuvant treatment; the database included more than 6500 patients undergoing pneumonectomy, 20% of which received induction. Cardiopulmonary complication and mortality rates were 21% and 5.2% after induction. Patients undergoing preoperative ChT combined with RT had a 2-fold higher mortality rate. After propensity score matching, the overall cardiopulmonary morbidity rate was significantly higher in pretreated patients compared to those undergoing upfront surgery.

In our study, the 5-year OS of patients undergoing pneumonectomy followed by adjuvant chemotherapy was apparently better than that of patients undergoing pneumonectomy without further treatment. Nevertheless, the OS curve of the non-adjuvant CT patients has a marked slope within the first 3 months after surgery; indeed, if we consider OS as starting 3 months after surgery, we see the difference between the two cohorts of patients vanishing.

Presumably, we may speculate that the worse OS recorded in non-adjuvant CT patients could be justified by the higher immediate postoperative death rate experienced by non-adjuvant treated patients.

In our institution, based on the previous literature data and on our own experience, all patients undergoing pneumonectomy with pN1 lymph node involvement are postoperatively evaluated by oncologists from the perspective of administering adjuvant chemotherapy. In fact, each patient with a pN1 disease after pneumonectomy should be a candidate for adjuvant CT. However, surgery-linked clinical conditions and postoperative complications, as well as comorbidities and overall clinical status, can influence this decision, leaning towards frequent follow-ups rather than adjuvant treatment. In our series, patients who were just followed up with had postoperative complications or deteriorating clinical conditions that prevented them from undergoing adjuvant CT. This consideration is further supported by the finding of overlapping curves between adjuvant CT and non-adjuvant CT patients in cancer-specific survival (CSS), meaning that the deaths within the first 3 months were not imputable to cancer progression, but to the surgery itself.

A selection bias is inherent in this analysis, since those undergoing adjuvant CT were, per definition, in better condition than those just being followed up with, but nonetheless, the survival of the two cohorts is comparable. The same could be observed after stratifying the patients per type of adjuvant treatment.

We have additionally reported the long-term follow-up of these patients, leading to a 10-year OS of more than 40% in patients undergoing pneumonectomy without neoadjuvant treatment, which is grossly overlapping the 42% reported by Riquet and colleagues [30] in 105 pN1 pneumonectomies. Nonetheless, our 10-year CSS is surprising: more than half of the patients who underwent pneumonectomy without pretreatment were still alive after 10 years, meaning that pneumonectomy, though burdened by significant morbidity and mortality rates, could heal selected lung cancer patients.

There are obviously some limitations to this study. First of all, its retrospective design and the long period investigated. In addition, an extensive preoperative pulmonary functionality was not included in the analysis, because cPET and pulmonary scintigraphy have only been routinely executed since 2017. Thus, the study of preoperative respiratory characteristics has not been uniform over time; nevertheless, exploring different features of this cohort of patients was not the aim of the study. We speculate that under the same conditions (pN1 candidates for adjuvant treatment), patients who were just followed up with experienced postoperative complications or conditions which did not allow them to be further treated, unlike those who underwent adjuvant treatment.

During follow-up, it could be useful to perform liquid biopsy, in order to dose ctDNA using blood or sputum samples. The analysis of liquid biopsy can provide continuous genetic mutation information during the time before or after adjuvant treatments, including its aggressiveness and overall molecular landscape.

Further studies focusing on patients undergoing pneumonectomies for pN1 NSCLC and adjuvant immune or target therapies are required.

Nonetheless, this is the largest real-world study investigating the early and long-term outcomes of pneumonectomy for pN1 NSCLC patients, comparing adjuvant treatment with follow-up.

## 5. Conclusions

In conclusion, patients with pN1 NSCLC completely resected by pneumonectomy and radical lymphadenectomy may represent a particular cohort which could be strictly followed up without adjuvant chemotherapy. In the era of targeting therapy and immunotherapy, this cohort may particularly benefit from these innovative therapies—since standard regimens have not been world changing—without chemotherapy utilization, taking into account its toxicity and its conflicting effects on overall survival.

## Figures and Tables

**Figure 1 cancers-16-03041-f001:**
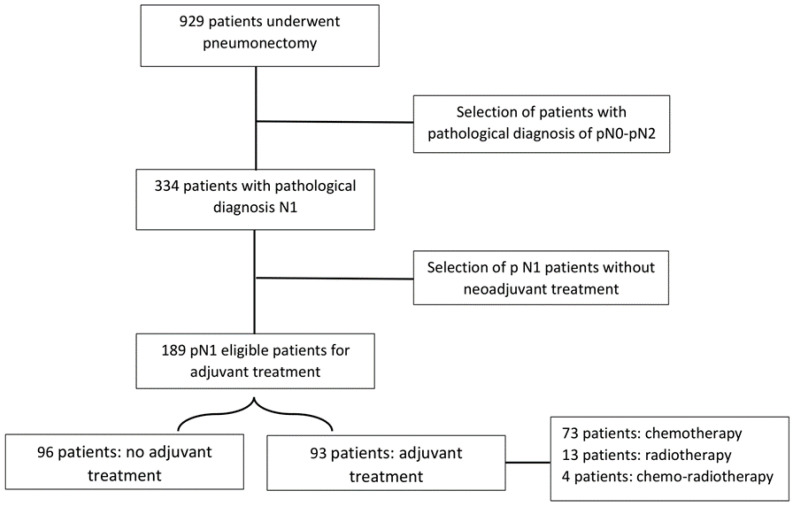
Study design.

**Figure 2 cancers-16-03041-f002:**
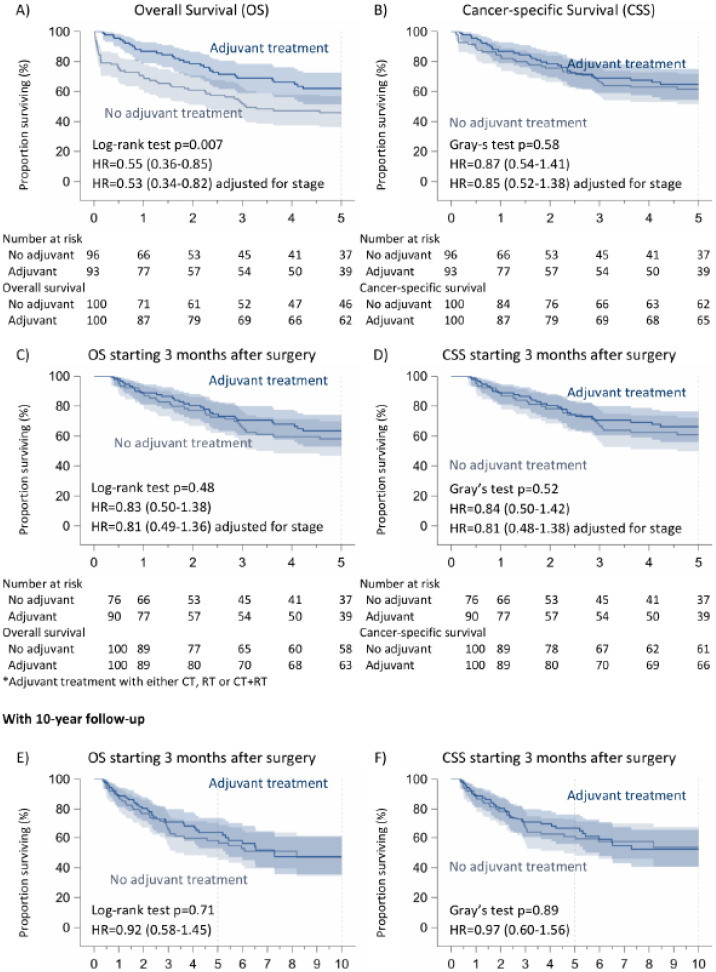
Overall and cancer-specific survival of non-pretreated patients with N1 NSCLC according to adjuvant treatment. (**A**) Overall Survival (OS), (**B**) Cancer-specific Survival (CSS), (**C**,**E**) OS starting 3 months after surgery, (**D**,**F**) CSS starting 3 months after surgery.

**Figure 3 cancers-16-03041-f003:**
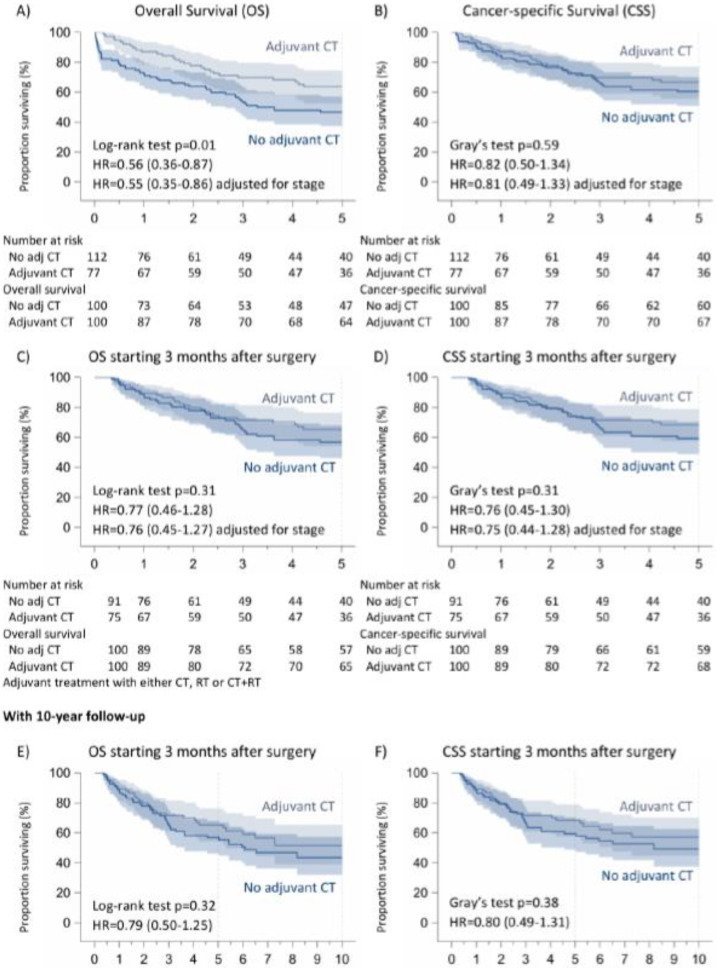
Overall and cancer-specific survival of non-pretreated patients with N1 NSCLC according to adjuvant CT. (**A**) Overall Survival (OS), (**B**) Cancer-specific Survival (CSS), (**C**,**E**) OS starting 3 months after surgery, (**D**,**F**) CSS starting 3 months after surgery.

**Figure 4 cancers-16-03041-f004:**
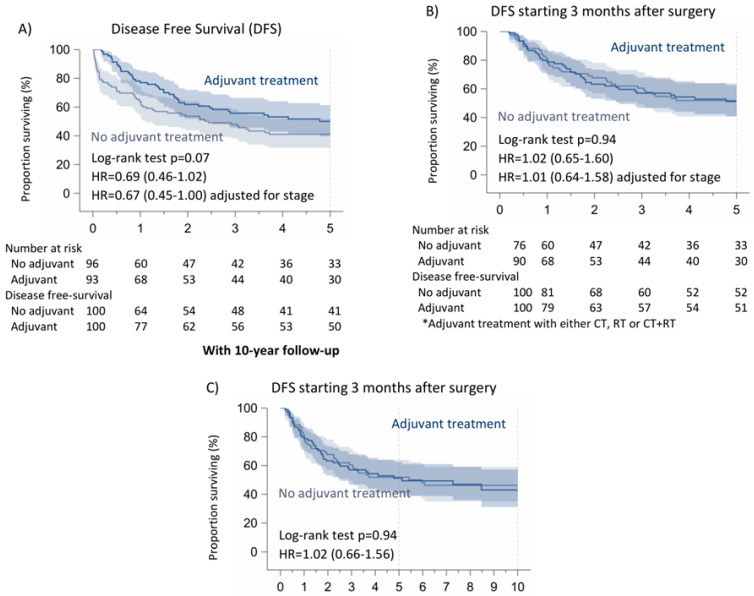
Disease-free survival of non-pretreated patients with N1 NSCLC according to adjuvant treatment. (**A**) Disease-Free Survival (DFS), (**B**,**C**) DFS starting 3 months after surgery.

**Figure 5 cancers-16-03041-f005:**
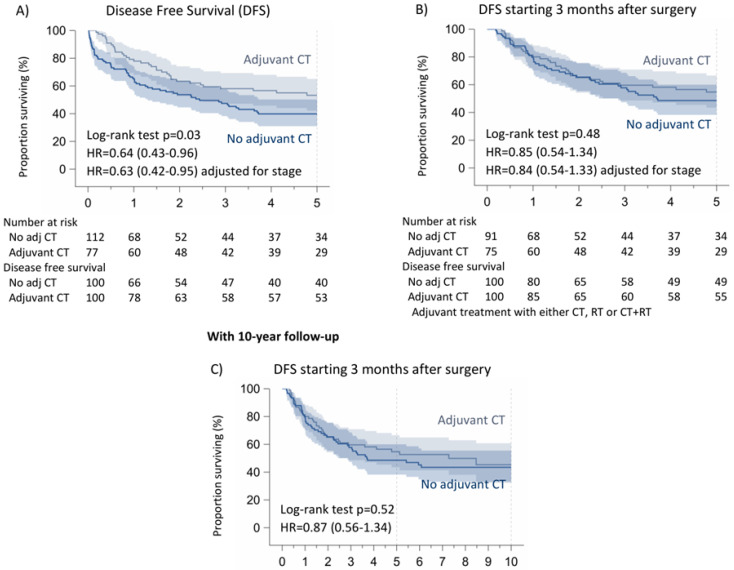
Disease-free survival of non-pretreated patients with N1 NSCLC according to adjuvant CT. (**A**) Disease-Free Survival (DFS), (**B**,**C**) DFS starting 3 months after surgery.

**Figure 6 cancers-16-03041-f006:**
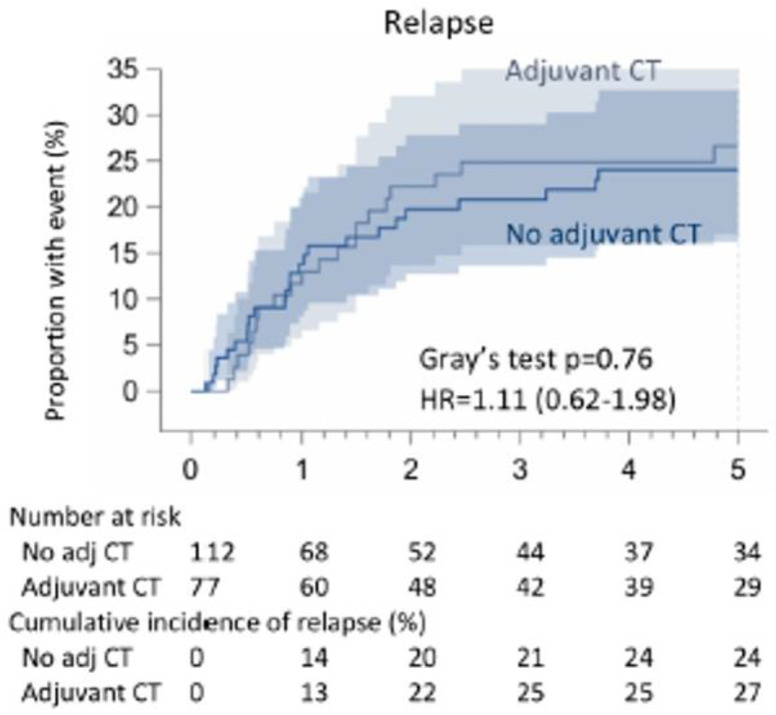
Cumulative incidence of relapse in non-pretreated patients with N1 NSCLC according to adjuvant CT.

**Table 1 cancers-16-03041-t001:** Characteristics of patients with N1 NSCLC.

	All	Neo Adjuvant	Naive				
		Total	Total	No Adjuvant	Adjuvant CT	Adjuvant RT	Adjuvant CT/RT
All	334	145	189	96	73	16	4
Age group							
<60	94	48	46	15	22	5	4
60–69	152	65	87	41	39	7	.
70+	88	32	56	40	12	4	.
Sex							
Male	264	111	153	83	56	10	4
Female	70	34	36	13	17	6	.
Cardiovascular comorbidity							
No	210	91	119	56	47	12	4
Yes	122	54	68	39	25	4	.
Unknown	2	.	2	1	1	.	.
Pulmonary comorbidity							
No	278	116	162	77	67	14	4
Yes	54	29	25	18	5	2	.
Unknown	2	.	2	1	1	.	.
Previous cancer							
No	283	122	161	76	68	13	4
Yes	51	23	28	20	5	3	.
BMI							
Normal weight	146	61	85	42	33	7	3
Over weight	130	61	69	39	21	8	1
Obese	32	12	20	10	9	1	.
Unknown	26	11	15	5	10	.	.
Histology							
NSCLC	3	1	2	1	1	.	.
Squamous	205	84	121	64	47	8	2
Adenocarcinoma	116	53	63	29	25	7	2
BAC	1	1	.	.	.	.	.
Adenosquamous	9	6	3	2	.	1	.
Stage							
Stage II	92	.	92	53	35	4	.
Stage III	97	.	97	43	38	12	4
Pretreated	145	145	.	.	.	.	.
Side							
Right	149	71	78	42	31	5	.
Left	185	74	111	54	42	11	4
Extensive resection							
No	235	84	151	72	65	11	3
Yes	99	61	38	24	8	5	1
Sleeve							
No	315	133	182	92	70	16	4
Yes	19	12	7	4	3	.	.
ICU							
None	124	53	71	32	34	3	2
1 day	163	70	93	45	34	13	1
2–5 days	31	15	16	10	5	.	1
>5 days	16	7	9	9	.	.	.
FEV1P, mean	80.9	80.2	81.5	80.8	82.1	85.2	73.9
DLCOP, mean	83.2	79.3	86.0	84.8	89.9	79.8	69.0

**Table 2 cancers-16-03041-t002:** Patients who died within 30 days or within 3 months of surgery.

	Within 30 Days of Surgery		Within 3 Months of Surgery	
			NO CT	CT
All	13		20	2
Age group				
<60	0		2	0
60–69	6		7	1
70+	7		11	1
Sex				
Male	13		20	2
Female	0		0	0
Cardiovascular comorbidity				
No	5		9	1
Yes	8		11	1
Pulmonary comorbidity				
No	8		13	2
Yes	5		7	0
Previous cancer				
No	1		16	2
Yes	2		4	0
BMI				
Normal weight	4		9	2
Over weight	8		10	0
Histology				
NSCLC			1	0
Squamous	9		13	1
Adenocarcinoma	4		6	1
Stage				
Stage II	6		7	1
Stage III	7		13	1
Side				
Right	10		15	1
Left	3		5	1
Extensive resection				
No	9		13	1
Yes	4		7	1
Sleeve				
No	12		18	2
Yes	1		2	0
ICU				
None	2		3	1
1 day	3		5	0
2–5 days	2		4	1
>5 days	6		8	0
FEV1P, mean	75.3		75.0	92.5
DLCOP, mean	88.9		83.6	70.0

## Data Availability

Data are contained within the article.

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
