# Peer review of "The Role of Adjuvant Chemotherapy in pN1 (IIB/IIIA) NSCLC Patients Who Undergo Pneumonectomy: Is It Still Justified in the Modern Era?"

_cancers, 2024, doi:10.3390/cancers16173041_

Round 1

Reviewer 1 Report

Comments and Suggestions for Authors

The article has valuable case resources, however, the design lacks rationality and the conclusions lack credibility. In the era of precision therapy, the selection of comprehensive treatments should account for the intricate pathology and genetic mutation disparities observed in tumors. But this article ignores both issues. In addition, ctDNA should be mentioned in both the introduction and discussion sections.

Comments on the Quality of English Language

Minor editing of English language required.

Author Response

The article has valuable case resources, however, the design lacks rationality and the conclusions lack credibility. In the era of precision therapy, the selection of comprehensive treatments should account for the intricate pathology and genetic mutation disparities observed in tumors. But this article ignores both issues. In addition, ctDNA should be mentioned in both the introduction and discussion sections.

I thank the author for his comments.

The aim of the paper was to evaluate chemotherapy alone in this particular population (N1 pneumonectomised), leaving out the entire molecular aspect, precisely to underline its role in the current world, increasingly characterized by molecular and precision medicine.

At present, for example, immunotherapy is administrated following chemotherapy; while in target therapies, CT has already been supplanted.

The aim was precisely to understand its real usefulness in such a particular population.

For this reason we have expressly excluded this aspect, describing however (section: materials and methods) our behaviour in the molecular study of patients.

We did not dose ctDNA in our preoperative setting or during follow-up.

Reviewer 2 Report

Comments and Suggestions for Authors

The paper evaluated the role of adjuvant chemotherapy in patients who underwent pneumonectomy for pN1 NSCLC. The research results have certain clinical value. However, the disadvantage is that the number of samples that meet the final screening criteria is still insufficient. The research method is appropriate, and the results are presented appropriately.

In short, the article was carefully prepared by the author.

I have no better suggestions.

Author Response

The paper evaluated the role of adjuvant chemotherapy in patients who underwent pneumonectomy for pN1 NSCLC. The research results have certain clinical value. However, the disadvantage is that the number of samples that meet the final screening criteria is still insufficient. The research method is appropriate, and the results are presented appropriately.

In short, the article was carefully prepared by the author.

I have no better suggestions.

I really thank the reviewer.

Reviewer 3 Report

Comments and Suggestions for Authors

The authors provide a 25 year experience from a single center on the use of adjuvant chemotherapy for patients NSCLC and pN1 disease after pneumonectomy. They provide a large cohort with adequate follow up. The authors propose that after excluding early deaths within the first 90 days postoperatively, adjuvant treatment with CT/CRT/RT does not affect overall or cancer specific survival. The study is well designed and is of great interest to the audience. Would recommend publication after minor suggestions.

-Table 1: given the large time span would describe the individual chemotherapeutic and immunotherapeutic agents used for adjuvant therapy

-Is there information on the interval between surgery and initiation of adjuvant chemotherapy? This can influence the efficacy of treatment.

-Along the same lines as above, how many patients completed the full intended course of chemotherapy, immunotherapy and/or radiation? Did this make a difference in outcomes?

-Line 165: would include the median follow up with interquartile range to place overall survival into perspective here

-Are postoperative functional status available? It appears that the main cause of the decision to not pursue adjuvant therapy was a global assessment of clinical status. Did outcomes change after adjusting for postoperative functional status?

-Table 2: Is the cause of death available for patients with an early death? It would be interesting to know whether these deaths are directly related to surgery, underestimated cancer burden, comorbidities, etc. This will allow the reader to better risk stratify these patients. The initial differences in survival likely reflect selection.

-Was there any difference in outcomes within specific N1 lymph node levels (more hilar vs peripheral)?

Minor:

-Lines 215-217: This paragraph seems to be out of place, perhaps from the author instructions  

Author Response

REVIEWER 3

The authors provide a 25 year experience from a single center on the use of adjuvant chemotherapy for patients NSCLC and pN1 disease after pneumonectomy. They provide a large cohort with adequate follow up. The authors propose that after excluding early deaths within the first 90 days postoperatively, adjuvant treatment with CT/CRT/RT does not affect overall or cancer specific survival. The study is well designed and is of great interest to the audience. Would recommend publication after minor suggestions.

-Table 1: given the large time span would describe the individual chemotherapeutic and immunotherapeutic agents used for adjuvant therapy

77 patients received adjuvant CT, consisting of doublet of platinum and pemetrexed if possible. We did not specifiy drugs for each patient, because some information was missing. We are a national reference center, but therapy often was administrated in centers closer to the patients residence; in some cases, these centers did not provide all the necessary information

-Is there information on the interval between surgery and initiation of adjuvant chemotherapy? This can influence the efficacy of treatment.

The timing for starting adjuvant therapy is 30-40 days maximum after surgery. We added into the text

-Along the same lines as above, how many patients completed the full intended course of chemotherapy, immunotherapy and/or radiation? Did this make a difference in outcomes?

All patients concluded at least two (out of three) lines of adjuvant CT. Mortality during CT was of 2 patients (table 2).

-Line 165: would include the median follow up with interquartile range to place overall survival into perspective here

Done

-Are postoperative functional status available? It appears that the main cause of the decision to not pursue adjuvant therapy was a global assessment of clinical status. Did outcomes change after adjusting for postoperative functional status?

This is an interesting remark. Immediate postoperative status assessement is mandatory for evaluating if adjuvant CT is tolerable or not. The immediate postoperative outcomes don’t normally improve in patients unfit for adjuvant treatments. When post-pneumonectomy complications (empyema, broncho-pleural fistula, cardiac complications) are resolved and the patients become fit again for adjuvant treatments, they  have left the “30/40 days after-surgery time window” for CT. In this case we decided to not administrated any drug.

-Table 2: Is the cause of death available for patients with an early death? It would be interesting to know whether these deaths are directly related to surgery, underestimated cancer burden, comorbidities, etc. This will allow the reader to better risk stratify these patients. The initial differences in survival likely reflect selection.

This is an interesting remark. We excluded patient with an early death (30-90 days after surgery) due to the close correlation with the surgical act (ARDS, respiratory failure, massive pulmonary embolism).  On the other hand, we know that pneumonectomy has a high mortality rate. However we did not focus on this aspect, because we did not consider it as the core of our work (we have published many works about  complications and mortality after pneumonectomy).

-Was there any difference in outcomes within specific N1 lymph node levels (more hilar vs peripheral)?

We didn’t massively explore the difference between hilar or peripheral lymph-node involvements. However, pneumonectomy represents an intervention required for its hilar pulmonary involvement.

Minor:

-Lines 215-217: This paragraph seems to be out of place, perhaps from the author instructions 

Thanks. Done